# Influence of Spanwise and Streamwise Film Hole Spacing on Adiabatic Film Effectiveness for Effusion-Cooled Gas Turbine Blades

**Matthew Courtis** [1,*] **, Alexander Murray** [1] **, Ben Coulton** [1] **, Peter Ireland** [1] **and Ignacio Mayo** [2]

[1] Oxford Thermofluids Institute, University of Oxford, Oxford OX2 0ES, UK; alexander.murray@eng.ox.ac.uk (A.M.); ben.coulton@eng.ox.ac.uk (B.C.); peter.ireland@eng.ox.ac.uk (P.I.)

[2] Rolls-Royce PLC, Derby DE24 8BJ, UK; mayo@rolls-royce.com

\* Correspondence: matthew.courtis@eng.ox.ac.uk or matthew.courtis.10@gmail.com

**Abstract:** To meet the challenges of increased thermal loads and performance demands on aero-engine turbine blades, more advanced cooling techniques are required. This study used a modification of the well-known Goldstein equation to predict film effectiveness for an individual film cooling hole and applied the Sellers' superposition method to apply these films across effusion-cooled configurations. In doing so, it tackles a relatively unchallenged problem of film holes in close spanwise proximity. An experimental set-up utilised infrared cameras to assess the film effectiveness of nine geometries of varying spanwise and streamwise spacings. Higher porosity led to increased thermal protection, and the spanwise spacing had the most profound impact, with film effectiveness approaching 0.9. Additionally, greater uniformity in the spanwise direction was observed. The modified Goldstein-Sellers method showed good agreement with experimental results although lateral mixing was underestimated. This method represents a tool that could be easily implemented in the industry for rapid assessment of novel cooling geometries.

**Keywords:** gas turbine; effusion cooling; heat transfer; turbine cooling; superposition

## 1. Introduction

Over the last several decades, the operating temperature of gas turbines has continued to increase by about 10 K per year. As a result of the second law of thermodynamics, an increase in temperature leads to an increase in overall engine efficiency. The consequence is high inlet temperatures seen by the turbine, imposing high thermal stresses, especially on the guide vanes and first several blade rows. To resolve such issues and maintain suitable dependability of engines, there is a need for turbine blade cooling systems. Major milestones, such as internal cooling and film cooling, have enabled inlet temperatures of up to 2000 K. The coolant flow rate should be minimised to limit aerodynamic losses, and consequently, as temperatures continue to move beyond 2000 K more advanced cooling methods are required. Increased engine efficiency will reduce fuel costs and, in turn, has the capability to reduce $CO_2$ emissions by over 180,000 kg per engine per year [1].

Transpiration cooling takes advantage of porous walls and injects coolant into the flow, creating a protective film between the surface wall and hot gas. Two main processes are involved to produce cooling:

1. Convective cooling—between the cooling fluid and porous wall;
2. Film cooling—a layer of coolant air over the blade reduces the convective heat flux from the hot gas cross flow,

Consequently, the study of turbine cooling performance generally takes two distinct forms: film effectiveness and metal effectiveness. The former represents the performance of only the films and is measured by how closely the adiabatic wall temperature can be kept to to the coolant temperature, as shown in Equation (1). Metal effectiveness combines

both the film cooling and internal convective cooling, providing a representation of the overall cooling performance.

$$\eta_f = \frac{T_\infty - T_{aw}}{T_\infty - T_c} \tag{1}$$

While transpiration cooling has been intensively researched for many years, such as early work produced in 1951 by NASA [2] or more recently by Polezhaev [3], successful application in engines has remained unattainable due to limitations of the porous materials available. Additionally, accurately modelling the aerothermal and mechanical stress fields remains a challenge as coupling is required between the flow, thermal and stress fields. These are made even more complex, given the shear number of film holes present in highly porous cooling schemes.

Despite this, there is a move from traditional film cooling methods to effusion cooling. Krewinkel [4] highlighted that the difference between transpiration and effusion cooling is not always clear, although the former tends to be associated with porous media and the latter with discrete film holes. Nevertheless, effusion cooling represents a path on the way to transpiration cooling that theoretically occurs when the solid and coolant reach thermal equilibrium. A significant number of studies on effusion cooling have been performed in recent years [5–9], which is unsurprising given the major benefits of film superposition. This occurs with reduced film hole spacing as a result of film jet interaction with the consequence of augmented film protection offered by each subsequent downstream film hole.

Superposition was first proposed by Sellers [10], who found that the summation of films from a single hole could be used to replicate multiple rows of films. This takes the form of Equation (2).

$$\eta_f(x,y) = 1 - \prod_{i=1}^{n}\Big(1 - \eta_{f,i}(x,y)\Big) \tag{2}$$

Experiments by Muska et al. [11] and more recently Murray et al. [12] reaffirm the superposition method and its usefulness in predicting multi-row film cooling. The latter used a modified Goldstein Equation [13] (described later, but in the form of Equation (3)) to predict the film distribution of an individual hole and, when combined with Sellers' superposition, found the additive effect of multiple film hole rows. Jiang et al. [14] found that even one additional staggered row (offset by half the primary spacing in both streamwise and spanwise directions, $x$ and $y$) produced much higher film effectiveness. Even at high injection velocities, where the coolant flow detached from the metal surface, the vortex pairs of the first cooling row pushed the second row films toward the surface resulting in quick reattachment and enhanced film protection.

Murray et al. [15] used the Sellers method as part of a decoupled solver (separately solving the internal and external thermal fields) and showed good agreement between experimental and numerical results, as well as an exhibit for efficient numerical studies for complex double-walled effusion systems. Despite these successes, no account was made for the proximity of adjacent holes (spanwise consideration) and, as such, were deemed to be independent. This causes an issue for more porous geometries, and most studies have focused on film cooling with relatively large spanwise distances. This study builds upon this work to incorporate the method to more porous geometries, which are required to move from effusion towards transpiration-like cooling.

The present study investigates the influence of streamwise and spanwise spacing on film effectiveness using a modified Goldstein–Sellers approach, which is explained in the next section. An infrared methodology measures the cooling performance for nine geometries at six coolant flow rates. Numerical results for varying film spacing are presented, and a more advanced cooling layout is explored. Experimental results are presented before a comparison between the two result sets are evaluated. The work

represents a preliminary step in understanding the importance of film hole proximity and extensions to the Goldstein–Sellers method.

## 2. Materials and Methods

### 2.1. Research Object

Film cooling performance was assessed using flat plate geometries, all of which had a staggered configuration of film holes inclined at 30° to the plate surface, as illustrated in Figure 1. The hole diameter in all cases was 4.0 mm, with additional dimensions also stated. Spacing between film holes was varied in both the streamwise and spanwise direction for $S_x/D = 3.0$, 4.5 and 6.0, and similarly for $S_y/D$, to produce a test matrix of nine geometries. The area of coverage of film holes was based on a 6×6 array for $S_y/D = S_x/D = 6.0$, and the other spacings had more rows and columns in order to meet the same coverage area.

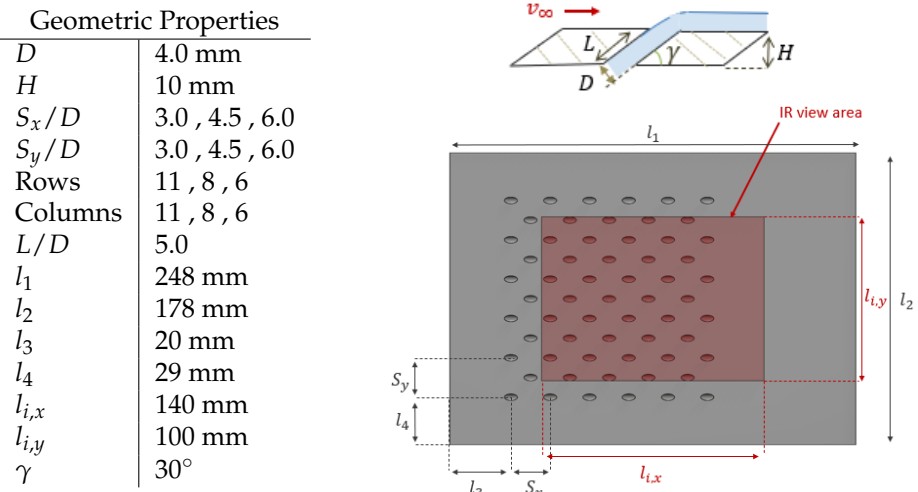

| Geometric Properties | |
|---|---|
| $D$ | 4.0 mm |
| $H$ | 10 mm |
| $S_x/D$ | 3.0 , 4.5 , 6.0 |
| $S_y/D$ | 3.0 , 4.5 , 6.0 |
| Rows | 11 , 8 , 6 |
| Columns | 11 , 8 , 6 |
| $L/D$ | 5.0 |
| $l_1$ | 248 mm |
| $l_2$ | 178 mm |
| $l_3$ | 20 mm |
| $l_4$ | 29 mm |
| $l_{i,x}$ | 140 mm |
| $l_{i,y}$ | 100 mm |
| $\gamma$ | 30° |

**Figure 1.** A schematic showing the dimensions of the flat plate geometry, where the red section highlights the visible portion for the IR camera. The illustration here shows the $S_x/D = S_y/D = 6.0$ geometry.

A more advanced film cooling system was also numerically investigated. A baseline design (TD2) was used and represents a transpiration-like cooling system with a porosity of 19%. This design was originally modelled on a double-wall effusion system developed by Murray et al. [16] at the Oxford Thermofluids Institute (OTI) (URL retrieved 1 September 2021) but with an increase in the number of film holes. The film hole diameter was also reduced to be 1/5 the size of conventional holes. The hole spacing was $S_x/D = S_y/D = 3.5$ with a hole diameter of 2 mm and a 'hole blockage' of 25%. The latter is defined as the ratio of film holes removed due to proximity to internal design features (pedestals and impingement jets). For direct comparison, the spanwise spacing was varied and the streamwise spacing altered to maintain porosity, as tabulated in Table 1. The film's effectiveness was found using the modified Goldstein–Sellers approach for each configuration at four blowing ratios ($M = 0.20, 0.25, 0.30, 0.35$) across the extent of the domain.

**Table 1.** Properties for the TD2 film spacing variation study.

| TD2 Study Geometric Properties | | | | | |
|---|---|---|---|---|---|
| $D$ | | | 2.0 mm | | |
| $S_x/D$ | 4.9 | 4.1 | 3.5 | 3.1 | 2.7 |
| $S_y/D$ | 2.5 | 3.0 | 3.5 | 4.0 | 4.5 |

## 2.2. Numerical Method

An in-house MATLAB code was used to predict the film effectiveness of any given film using a modified Goldstein correlation. This method attempts to model the decay of films in both the streamwise and spanwise directions and is defined in Equation (3). $M$ is the blowing ratio of the film, specified as the product of the ratios of density and velocity between the coolant and free-stream fluid, $M = (\rho_c v_c)/(\rho_\infty v_\infty)$. This parameter influences the following constants: turbulent thermal diffusivity $\alpha_t$ and $x_{decay}$, which captures changes in the streamwise direction. The spanwise distance constant ($c_1$) and the spanwise shaping constant ($c_2$), which influence the spanwise film decay, are variables dependent on both the blowing ratio $M$ and distance downstream of the film hole $x$. The values for these are based on experimental and correlation data, which can be found in [12,17].

$$\eta_f(x,y) = M\frac{V_\infty D}{8\alpha_t\left(x/D + x_{decay}\right)} \exp\left\{-\left(\frac{y}{c_1}\right)^{c_2}\right\} \qquad (3)$$

Films were placed individually on a global grid and subject to the Sellers' superposition, as illustrated in Figure 2. This method could take into account variable blowing ratios and any form of spatial configuration. It is worth noting that the empirical data used had a hole aspect ratio ($L/D$) of 2.0, compared to $L/D = 5.0$ for geometries investigated in this study. Lutum and Johnson [18] demonstrated that a lower aspect ratio reduced film effectiveness, and consequently, the film effectiveness calculated here may represent a conservative estimate of the actual value.

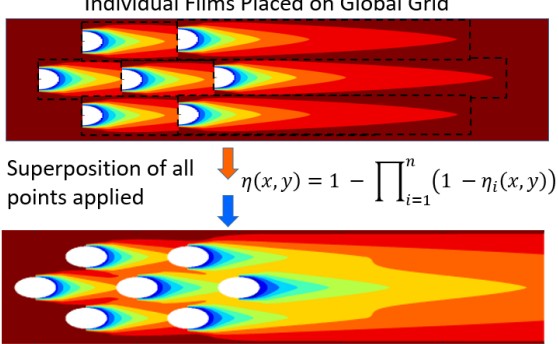

**Figure 2.** Visualisation of Sellers' superposition method.

## 2.3. Experimental Method

### 2.3.1. Experimental Facility

The **D**ouble-**W**alled **E**ffusion **C**ooling **A**erothermal **F**acility (DECAF) at the Oxford Thermofluids Institute was used for the experimental study—see Figure 3. The facility provides conditions for scaled tests of flat plate cooling geometries. Murray et al. [19] provides a detailed description of the facility and measurement procedure, although a brief summary is also provided here.

The facility provides Reynolds number similarity to engine conditions and has been specified for high-temperature capabilities. ROHACELL was used for test geometries to investigate film effectiveness, assuming an adiabatic surface. This material was chosen due to its low thermal conductivity ($\lambda \sim 0.030$ W/m-K), and a high grade with a closed cell structure was selected (71 HERO) to reduce the impact that roughness would have on film holes with small diameters. The film effectiveness is calculated by measuring the surface temperature through an infrared camera, and the back-side surface temperature is also monitored. A range of mass flows between 0.01 kg/s and 0.035 kg/s were investigated for each geometry. This was achieved by varying the coolant mass flow into the plenum while keeping the mainstream velocity constant. The typical experimental mainstream Reynolds number based on the film hole diameter, mainstream temperature, coolant-mainstream temperature ratio and density ratio are tabulated in Table 2.

**Table 2.** Approximate experimental conditions.

| $Re_\infty$ | $T_\infty$ | $T_c/T_\infty$ | $\rho_c/\rho_\infty$ |
|---|---|---|---|
| $1.12 \times 10^4$ | 373 K | 0.8 | 1.2 |

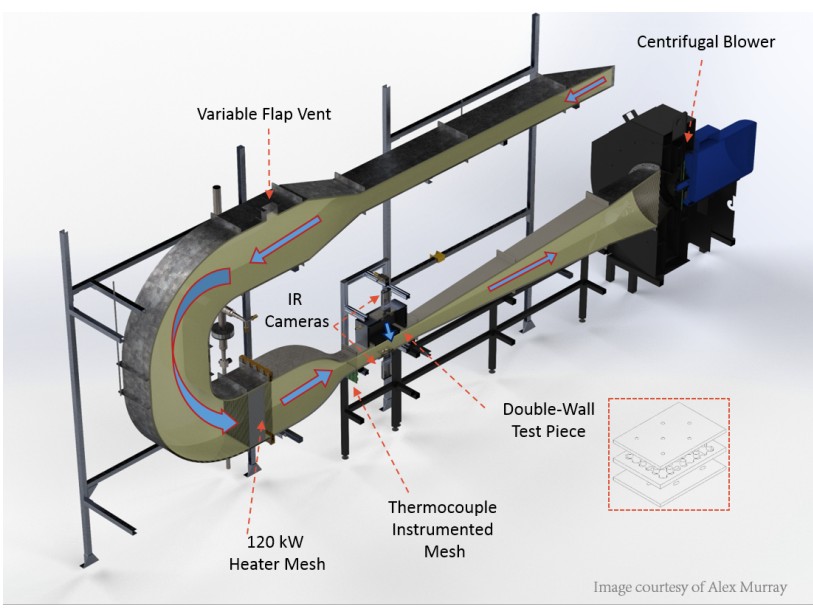

**Figure 3.** Annotated schematic of the OTI experimental facility.

### 2.3.2. Infrared Methodology

Temperatures of the external surface were made through the use of two *FLIR A655sc* infrared cameras, with optical access provided by Zinc Selenide windows. Thin wire thermocouples were attached to both the hot-gas and coolant plenum external surfaces of the test geometry. These are within the IR camera field of view, permitting an in situ calibration. External surfaces were painted matt black to minimise IR reflections from the surroundings. The calibration method is described in detail by Murray et al. [19], in which both the blackbody and greybody calibration were used to calculate temperatures from the raw data output of the cameras. The former finds the ideal blackbody parameters, and the latter finds parameters due to non-ideal surface IR emission. The frame rate of the camera was set at 25 Hz to allow an assessment of flow unsteadiness, and data are averaged over 10 s (250 frames) to provide a mean steady-state temperature. Using these temperatures, the film effectiveness was calculated using Equation (1).

## 3. Results and Discussion

### 3.1. Numerical Results

To determine the affect of changing streamwise ($S_x$) and spanwise ($S_y$) spacing on the film effectiveness, the modified Goldstein method with Sellers' superposition was implemented across a standardised geometry. The first study focused on maintaining a constant blowing ratio with three rows and three columns for a staggered array, which were varied as $3.0 \leq S_y/D \leq 5.0$ and $3.0 \leq S_x/D \leq 8.0$. A visualisation of this, along with the comparison area, can be seen in Figure 4. Another set of calculations were based on the assumption of constant mass flow between geometries. To achieve this, each geometry was made to cover the same surface area, and the coolant mass flow kept constant, as illustrated in Figure 5.

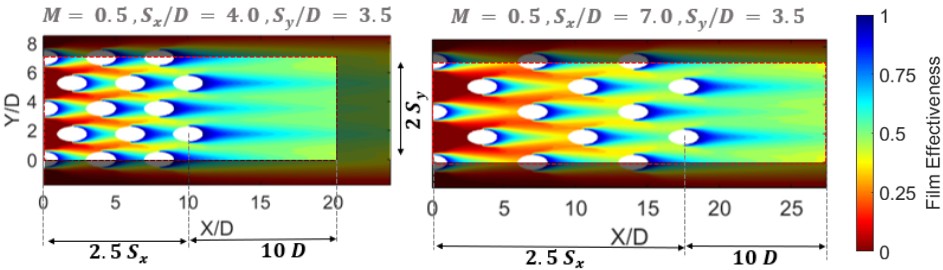

**Figure 4.** Comparison area for the mean film effectiveness with varying $S_x/D$ and $S_y/D$, using a constant blowing ratio. Enhanced superposition occurs with reduced streamwise spacing.

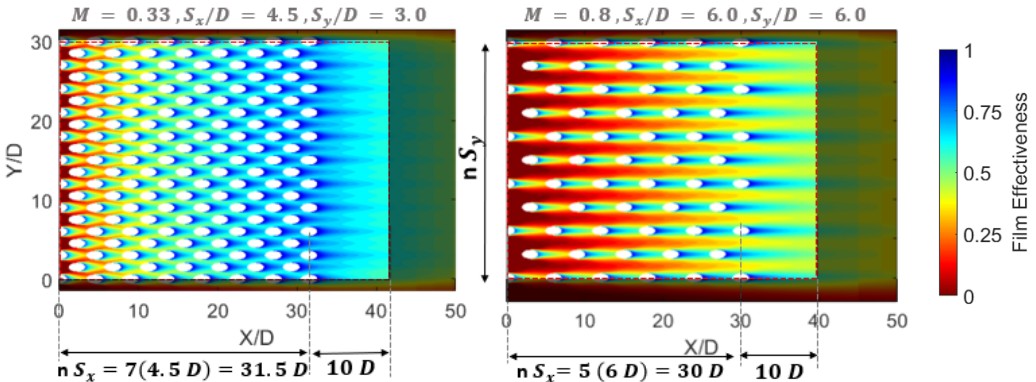

**Figure 5.** Comparison area for the mean film effectiveness with varying $S_x/D$ and $S_y/D$, with a constant coolant mass of $\dot{m}_c = 2.56\,\text{kg/s/m}^2$. Hot streaks are minimised as the spanwise spacing reduces.

For a fixed mass flow rate, increasing the porosity reduces the blowing ratio. While this reduces the performance of any individual film, the proximity of neighbouring holes may compensate for this detriment. Figure 5 illustrates that as spanwise distance increases, the films become more isolated, and distinct hot-streaks can be observed between film columns. As $S_y$ decreases, the extent of the hot streaks is reduced given the adjacency of a staggered film, and less of the exterior wall is subject to the mainstream temperature, a benefit even without accounting for film interaction. Additionally, the lower spanwise distance resulted in the merging of the adjacent films. Film effectiveness is enhanced by reducing $S_x$, as would be predicted by Sellers superposition, but since films propagate with the mainstream flow, the proximity in the streamwise direction is less influential than that for the spanwise direction.

The averaged film effectiveness for the first study is compared to porosity in Figure 6, where the symbols correspond to a constant $S_y$. Here, porosity is defined as the geometric porosity, the surface area of the film holes ($A_h$) compared to the area these holes cover ($A_s$), and is defined in Equation (4).

$$\phi = \frac{A_h}{A_s} = \frac{\pi}{2(S_x/D)(S_y/D)\sin(\gamma)} \tag{4}$$

As spanwise spacing was reduced, the area-averaged film effectiveness increased at a greater rate than the influence of streamwise spacing. This is made clear on the figure by two arrows indicating the general trend of film effectiveness for a reduction in $S_y$ and $S_x$, respectively. Consequently, for an increase in porosity and to reduce the added mass flow required, having films with a lower $S_y$ was advantageous.

To investigate constant mass flow across all geometries, the area-averaged film effectiveness is shown in Figure 7, alongside isobars of constant porosity. As one moves along a constant porosity curve, the film effectiveness benefits from decreasing $S_y$ and increasing

$S_x$. This was true for all three mass flow rates, confirming that increasing the porosity will enhance the overall film effectiveness.

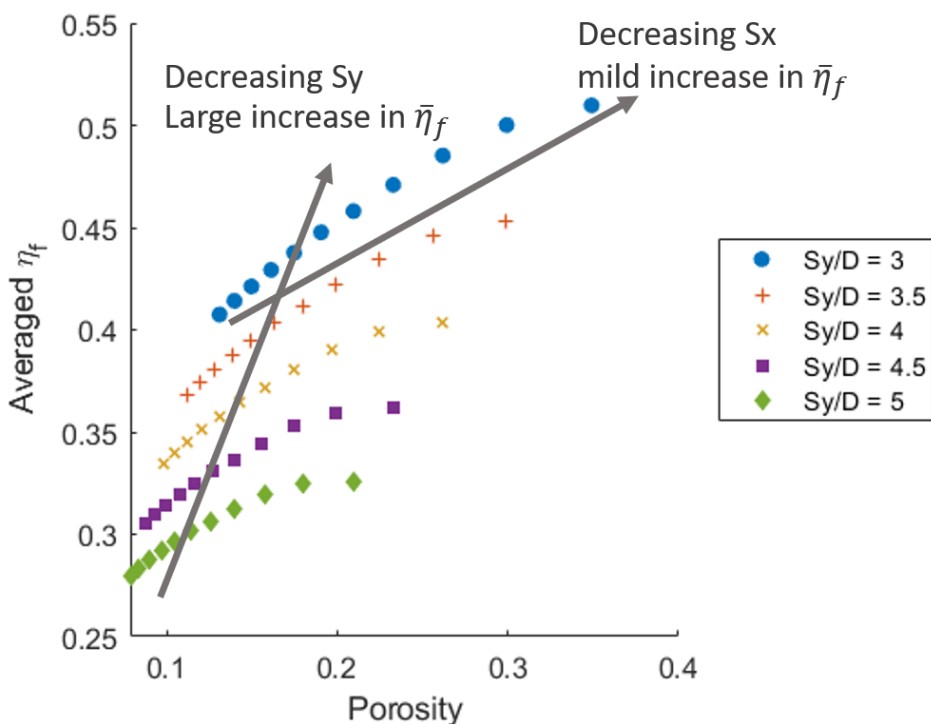

**Figure 6.** The effect of film spacing ($S_x$,$S_y$) on area-averaged film effectiveness, for $M = 0.5$

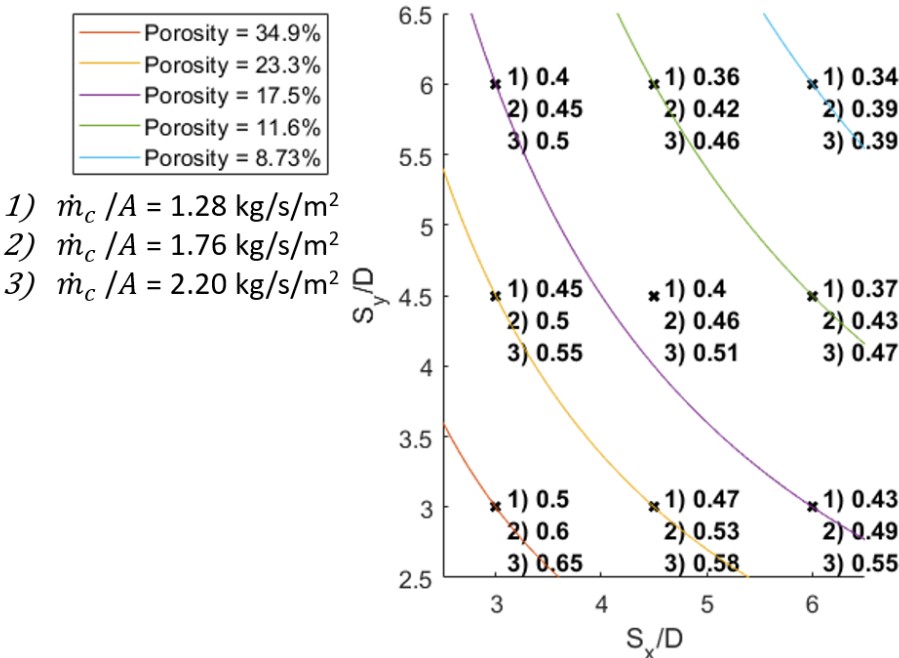

**Figure 7.** The effect of film spacing ($S_x$,$S_y$) on area-averaged film effectiveness, for three coolant mass flows. Isobars of constant porosity are displayed.

### 3.2. Td2 Results

High film effectiveness was demonstrated by TD2, as well as the design variants. Figure 8 plots the area-averaged film effectiveness of each variant characterised by the

ratio of streamwise to spanwise spacing ($S_x / S_y$) for four blowing ratios. The same trend of increased film effectiveness with a reduction in $S_y$ is observed. Increasing the blowing ratio increased the film effectiveness throughout, although with diminishing returns for each blowing ratio increment. Film effectiveness contours for two TD2 variants and the base geometry are shown in Figure 9 at $M = 0.3$. It can be seen that the leftmost plot (smallest $S_y$) has increased spanwise uniformity and the reduction of hot spots observed at the location where film holes were removed. The increased lateral spread compensates for a missing film hole and maintains the uniformity of film coverage. This is consistent with results by Yang et al. [20], which demonstrated that while the blockage ratio was an important parameter affecting film effectiveness for a porous metal plate, ratios of up to 30% had only a small influence.

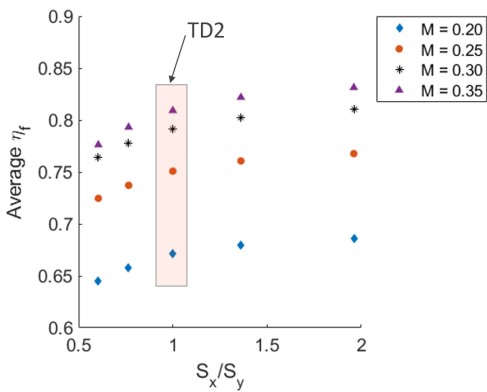

**Figure 8.** The area-averaged film effectiveness is compared to the ratio of spanwise and streamwise spacing for TD2-base geometry for four blowing ratios. The original design with $S_x = S_y = 3.5D$ is highlighted for reference.

For streamwise columns without hole removal (the second column from the bottom and for figures as we move left to right at $Y/D = 2.5$, $Y/D = 3.5$ and $Y/D = 4.5$), the increased streamwise spacing reduced the superposition effect, but given these are well protected areas and reach saturation after several rows, one would expect this not to be a concern for the overall blade design. These results suggest that further double-walled effusion designs should consider implementing low spanwise spacing if feasible.

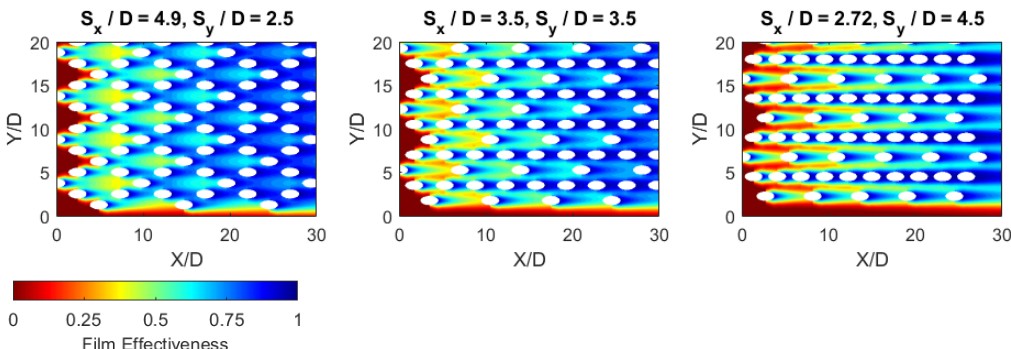

**Figure 9.** Film effectiveness contours for TD2 and two film spacing variations, for $M = 0.30$. TD2 ($S_x = S_y = 3.5$) is the central plot.

### 3.3. Experimental Results

To account for differences in tunnel velocity and fluid properties between experimental runs, a non-dimensional mass flow was used for comparison, defined in Equation (5), where $\bar{h}$ is the mean heat transfer coefficient (W/m$^2$/K) and $A$ is the surface area of the flat plate.

$$m^* = \frac{\dot{m}_c c_p}{\bar{h} A} \tag{5}$$

The averaged film-effectiveness for the visible camera area was calculated for each condition and is displayed in Figure 10. At $m^*$ near 1, there are three distinct groupings of film effectiveness, each with constant $S_y$. Between the three groups, as $S_y$ decreases, the film effectiveness increases. For larger values of $m^*$, these lines diverge, and the porosity (i.e., $S_x$) determines the film effectiveness range. For clarity, the lowest spanwise spacing results ($S_y/D = 3.0$) are highlighted and exhibited the best performance, across the entire range of $m^*$. This indicates that $S_y$ is more crucial to film protection than $S_x$. Furthermore, the line plotted with triangles ($S_x/D = 4.5$, $S_y/D = 6.0$) and stars ($S_x/D = 6.0$, $S_y/D = 4.5$) can be seen when comparing two geometries with the same porosity. Across the entire range of $m^*$, the latter geometry, with a lower $S_y$, has a higher average film-effectiveness.

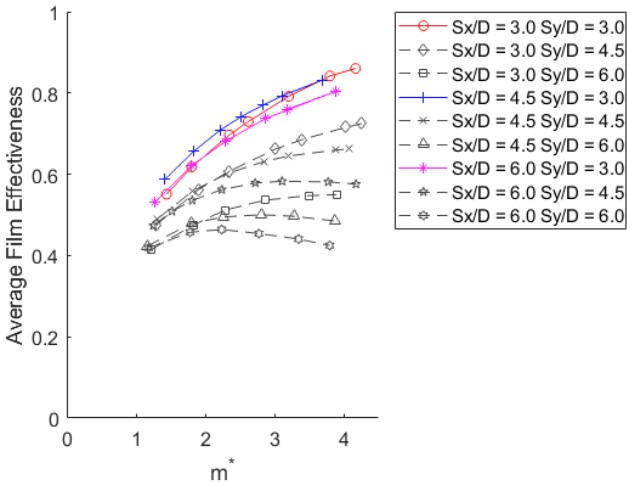

**Figure 10.** Experimental overall film effectiveness against non-dimensional mass flow ($m^*$). Geometries with $S_y/D = 3.0$ are highlighted and have the highest averaged film protection, illustrating the benefits associated with low spanwise film spacing.

The general characteristic of these curves is an increase in film protection with coolant flow, which plateaus as the coolant saturates, and for $S_y = 3.0$ geometries, they approach a film effectiveness of 0.9. For lower porosity cases, not only is a plateau reached, but the film effectiveness reduces. This is a consequence of high blowing ratios, and the occurrence of jet lift-off. An unexpected observation was a higher average film effectiveness for $S_x/D = 4.5$, $S_y/D = 3.0$ compared to the most porous geometry ($S_x/D = S_y/D = 3.0$) for all but the highest mass flow, discussed in more detail below.

Contours of film effectiveness for two mass flows, approximately $m^* = 1.8$ and $m^* = 3.2$, are illustrated in Figures 11 and 12, respectively. Enhanced film protection is evident for the most porous geometries, as well as the benefit of low spanwise spacing. The most striking feature is that film hole proximity to adjacent ('staggered') rows enhances the superposition of films. This is still present at $S_y/D = 4.5$, albeit to a lower extent. However, for $S_y/D = 6.0$, it is almost absent.

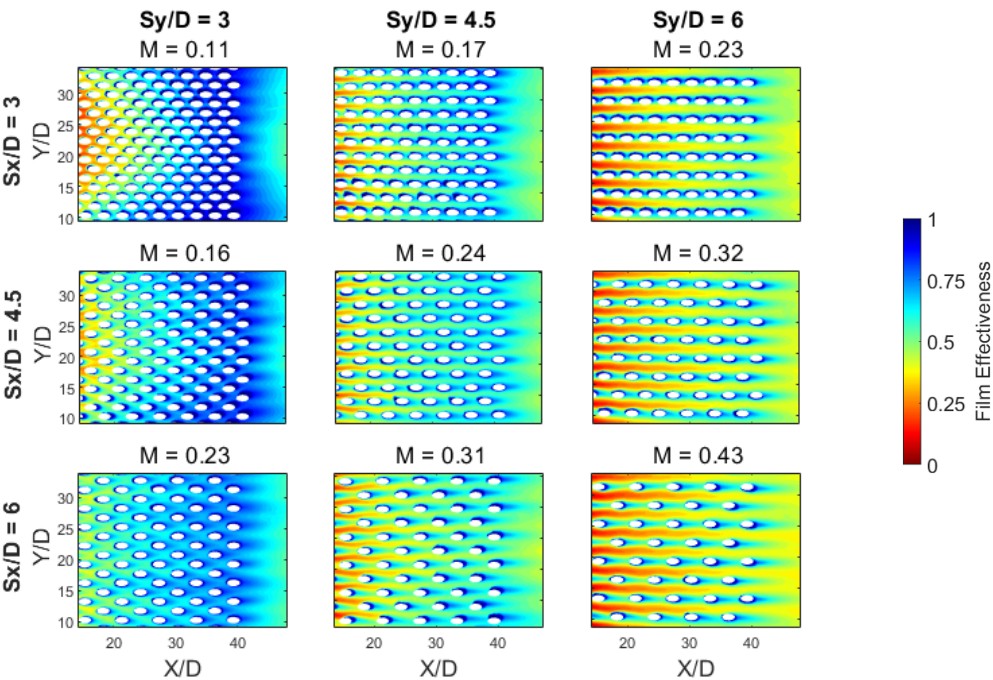

**Figure 11.** Experimental film effectiveness contours for $m^* \approx 1.8$. The blowing ratio ($M$) of each geometry is displayed above each contour plot. In general, increased porosity results in a more uniform film coverage.

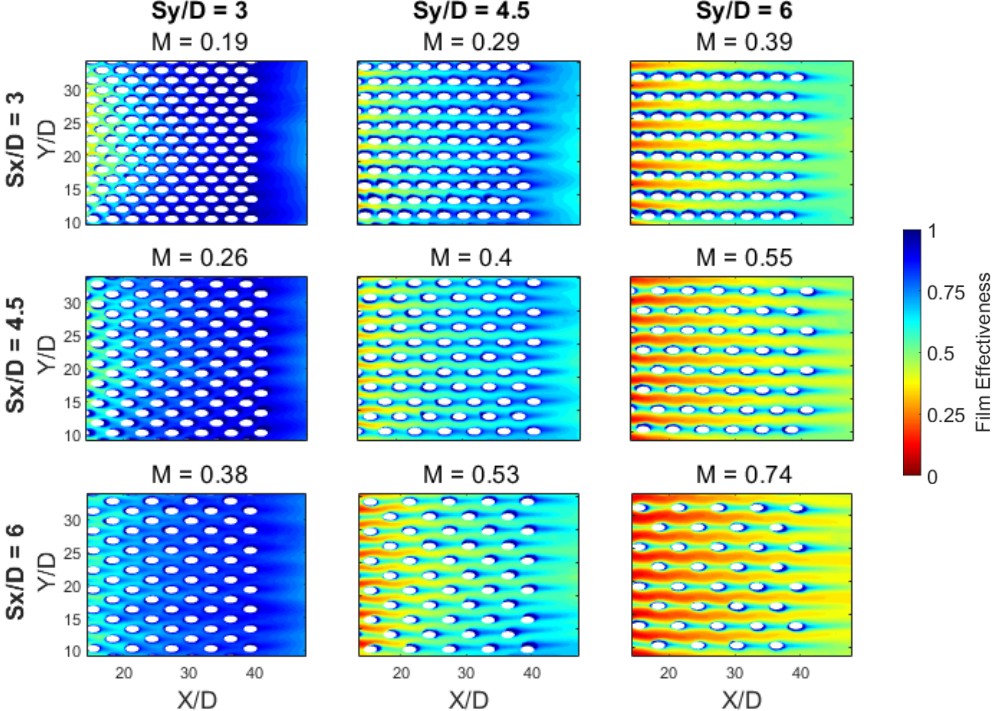

**Figure 12.** Experimental film effectiveness contours for $m^* \approx 3.2$. The blowing ratio ($M$) of each geometry is displayed above each contour plot. High porosity geometries form saturated films, leading to uniform and almost complete protection. For lower hole counts, higher blowing ratio leads to jet liftoff, reducing film effectiveness.

At the two highest porosities with $S_y/D = 3.0$ ($S_x/D$ of 3.0 and 4.5), there is a variation in film effectiveness across the spanwise direction. A similar feature was observed

by Murray et al. [12], where a flat plate with hole spacings of $S_x/D = S_y/D = 3.0$ was investigated using pressure-sensitive paint for a range of blowing ratios. It was suggested to be a result of variations in the blowing ratio between film holes across the span, and a product of the side-wall boundary layer reducing the mainstream flow velocity in the vicinity of the side film holes. In this study, given the distance between the side walls and the test piece, another explanation for the variation in blowing ratio is provided here. At large spanwise spacings, the mainstream flow can migrate around individual film holes, evidenced by hot streaks. However, as $S_y$ decreases, the mainstream flow must also divert around the nearby staggered holes. Consequently, the increased upwards curvature of the mainstream flow imposes an increased static pressure on the film hole outlet. Given the finite lateral boundary of the cooling array, film holes at the edge do not pose as large of a obstacle for the mainstream flow, as it can curve both upwards and laterally; thus, the static pressure is lower at the edge of the cooling array compared to the middle. This explains the higher averaged film effectiveness for $S_x/D = 4.5, S_y/D = 3.0$ compared to $S_x/D = S_y/D = 3.0$, as the coolant deviates to the periphery holes rather than those that are central. Additionally, the upstream central holes for the latter geometry may struggle due to the lower blowing ratio, which may lead to hot-gas ingestion at the leading central film holes. Despite this feature, geometries with $Sy = 3.0$ still exhibit a high film effectiveness throughout the visible domain, even along the centreline..

The streamwise-averaged film effectiveness is introduced here and shown in Figures 13 and 14, with a constant $S_y$ for each plot. This metric is an indication of how uniform the protection is perpendicular to the mainstream flow. For film holes spaced with $S_y/D = 6.0$, the difference between the peak and trough streamwise-average film effectiveness is approximately 0.4, compared to 0.2 for $S_y/D = 4.5$ and 0.15 for $S_y/D = 3.0$. This highlights that a reduction in $S_y$ increases the mean film effectiveness, as well as reducing the variance. Hence one can conclude that if a uniform protection is desired, the spanwise film spacing is crucial to this aim. On the contrary streamwise spacing has only a small influence on this but acts to increase the mean film effectiveness.

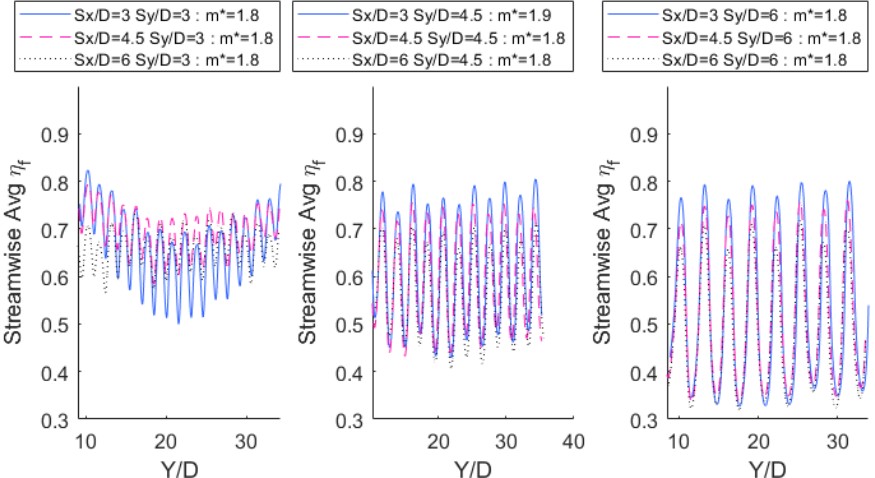

**Figure 13.** Streamwise-average film effectiveness for $m^* \approx 1.8$. In each individual plot, geometries with constant $S_y$ are shown.

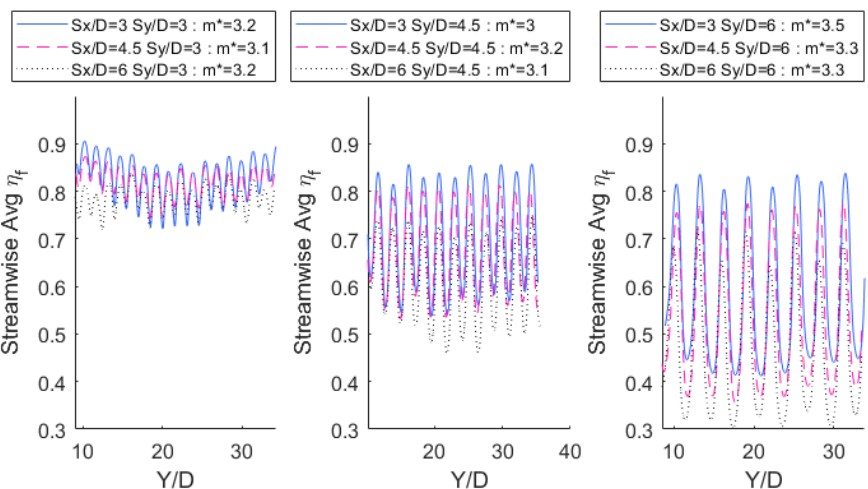

**Figure 14.** Streamwise-average film effectiveness for $m^* \approx 3.2$. In each individual plot, geometries with constant $S_y$ are shown.

### 3.4. Numerical and Experimental Comparison

Spanwise and streamwise averaged film effectiveness were calculated using the modified Goldstein–Sellers across the same area as the infrared camera: $14D < x < 48D$ and $9D < y < 34D$. Results for $m^* = 1.8$ are shown in Figures 15 and 16 and results for $m^* = 3.8$ by Figures 17 and 18, with the solid and dashed line representing experimental and numerical results, respectively.

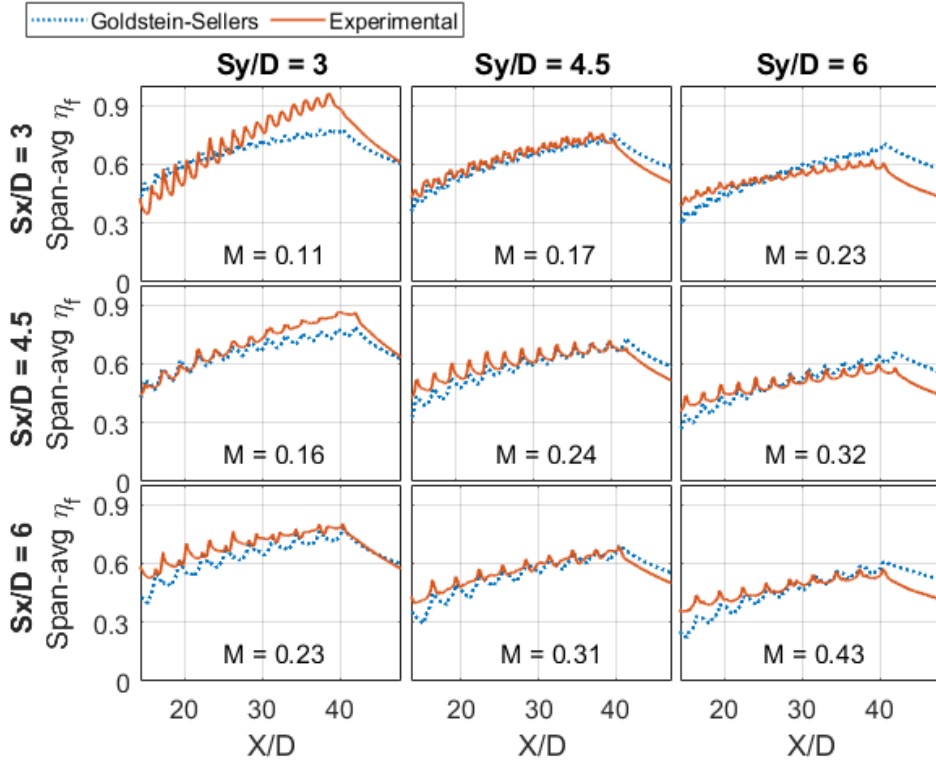

**Figure 15.** A comparison of numerical and experimental spanwise-averaged film effectiveness for $m^* = 1.8$.

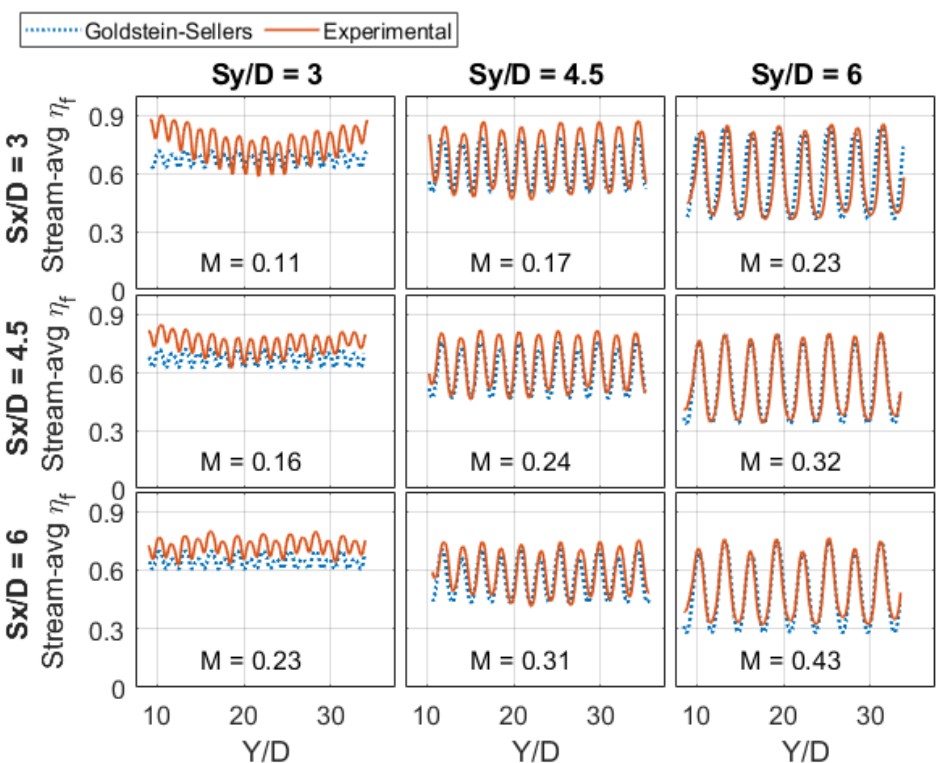

**Figure 16.** A comparison of numerical and experimental streamwise-averaged film effectiveness for $m^* = 1.8$.

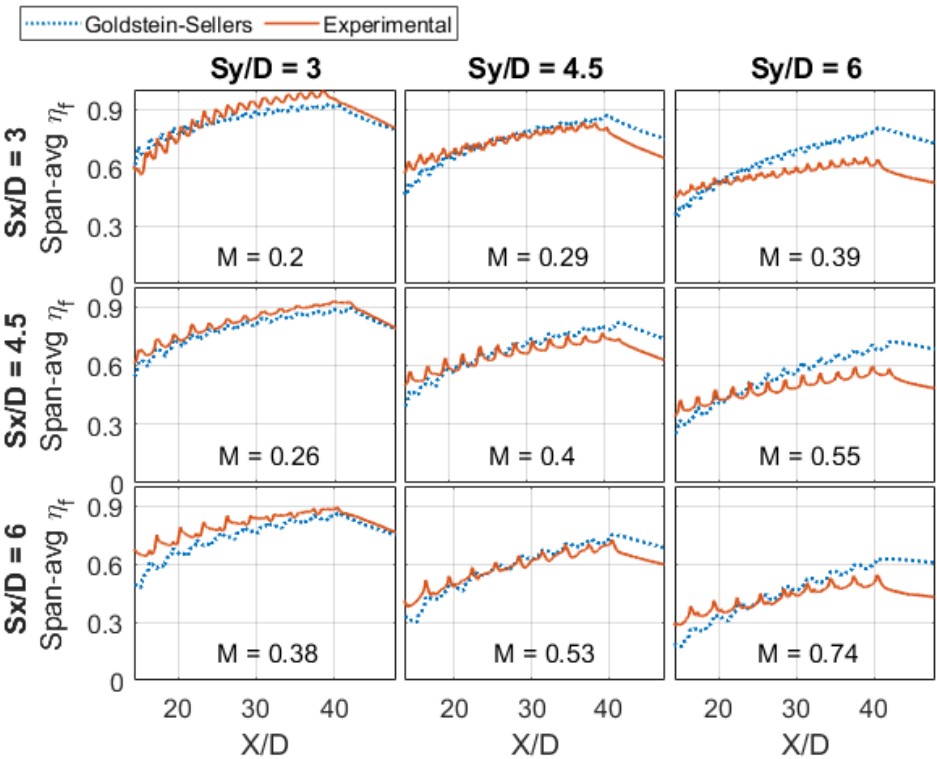

**Figure 17.** A comparison of numerical and experimental spanwise-averaged film effectiveness for $m^* = 3.8$.

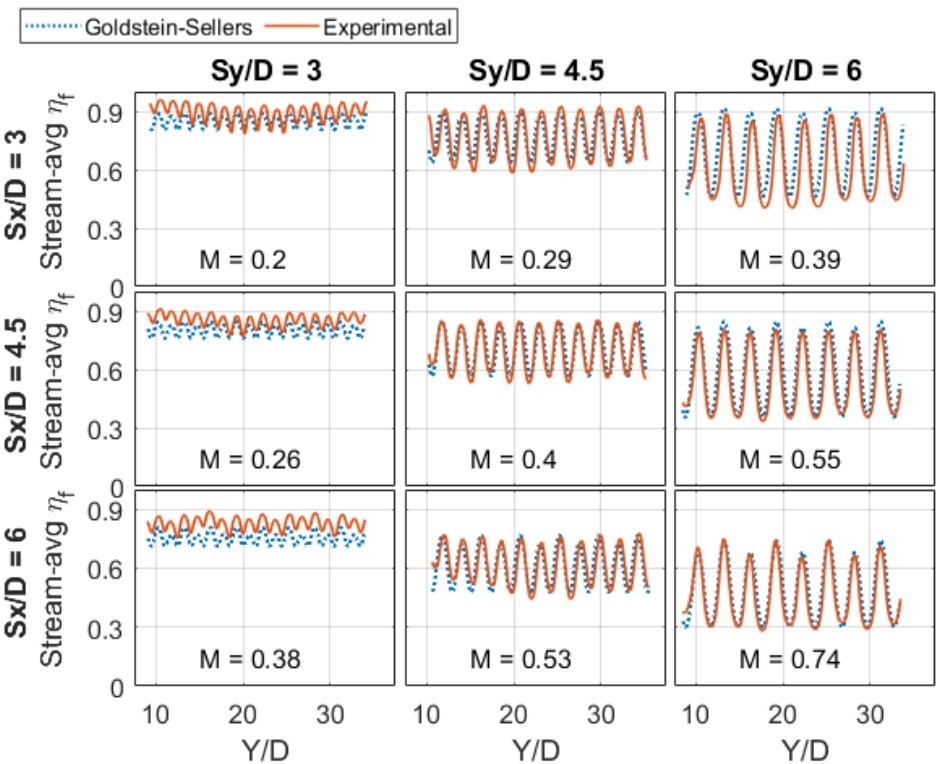

**Figure 18.** A comparison of numerical and experimental streamwise-averaged film effectiveness for $m^* = 3.8$.

The modified Goldstein–Sellers captures the general trend of the spanwise-average film effectiveness across all geometries. For the largest spanwise spacing ($S_y = 6D$), the spanwise-average $\eta_f$ is under-predicted for the leading film holes up to $X/D = 20$ but over-predicted beyond $x/D = 40$, indicating that while the initial film protection was underestimated, the effect of superposition was overly optimistic. The same trend can be observed for $S_y = 4.5D$ but to a lesser extent. For the lowest spanwise spacing ($S_y = 3D$), spanwise-averaged $\eta_f$ was well captured despite a small under-prediction throughout the domain, with a deviation for $S_x = 3D$, $S_y = 3D$ due to the suspected variable mass flow distribution, as previously discussed.

The streamwise-average film effectiveness indicates that films are well-matched for $S_y = 6D$ and $S_y = 4.5D$, with numerical results having a higher film effectiveness directly downstream of the film holes and a lower film effectiveness in the gaps between film hole columns. This would suggest a lack of lateral spread film coverage was predicted numerically. For $S_y = 3D$, the numerical model under-predicts throughout the plate; however, the level of variation between film rows was similar.

Spanwise and streamwise averaged film effectiveness indicate that the modified Goldstein–Sellers method can predict the superposition of staggered film arrays for both densely packed and moderately spaced film holes. This is further examined by the film effectiveness contours for both experiments and the numerical method in Figure 19a,b. Three geometries with evenly spaced film holes ($S_x = S_y$) are shown at two mass flows, with blowing ratios indicated above each plot. For the lowest film spacing ($S_x = S_y = 3D$), the two mass flows paint a contrasting picture. At $m^* = 1.8$, the experiments had lower film effectiveness at upstream film rows, and this was more pronounced at the centre-line of the array. Conversely, film effectiveness builds up more as you move downstream compared to Goldstein–Sellers. This was largely due to the variable mass flow distribution observed for highly porous geometries. As mass flow increases to $m^* = 3.8$, this phenomenon was less pronounced and may be due to earlier saturation of film effectiveness, and thus, differences are constrained to upstream rows outside the IR camera view. The authors suggest that

investigation of the upstream rows for densely packed film arrays should be made in future work.

**Figure 19.** A comparison of film effectiveness contours between experiments and Goldstein–Sellers. Film hole spacing is stated for each row, and the corresponding blowing ratios are displayed above each plot.

For the two other geometries shown in Figure 19 ($S_x = S_y = 4.5D$ and $S_x = S_y = 6D$), film effectiveness was over-predicted for film hole centre-lines; however, this appears to be at the expense of hot streaks between film holes. This is an indication of increased mixing in the experimental study that is not accounted for by the Sellers approach. Some of the discrepancy may be due to manufacturing differences between the experimental pieces in this study compared to that used by [12,17] in the formulation of the modified Goldstein equation. In this study, the flat plates were made from ROHACELL, and consequently, the hole boundaries had significant deviations that may enhance the lateral spread of films and increase the turbulence of the film jets.

## 4. Conclusions

The effect of variable spanwise and streamwise film hole spacing was investigated for staggered hole arrays. While a decrease in both spacings enhances the film effectiveness, the spanwise spacing is more effective, and jet lift off at higher mass flows can be avoided. Additionally, the range of streamwise-averaged film effectiveness is reduced by half when moving from $S_y = 6.0$ to $S_y = 4.5$, indicating a more uniform coverage. It is thus worth considering moving away from equal spanwise and streamwise spacing for effusion designs, despite this being common practice in the industry. This could be important in designs looking to increase film hole porosity, as manufacturing realities may introduce the need for missed rows for structural purposes. Further work needs to address variable mass flow distribution between film holes for geometries with porosity in excess of 20%.

Highly porous film-hole coverage represents an efficient cooling method with film effectiveness approaching 0.9. However, metal effectiveness and mechanical stresses should be considered for a better understanding of the overall consequences on more engine-like designs. This is an area of research investigated by Elmukashfi et al. [21], and something that the authors of this study look to expand upon.

A modified Goldstein–Sellers film superposition method was compared to experimental results for a range of staggered film hole spacings. This method was shown to predict the behaviour of film superposition relatively well but did not completely account for lateral mixing. Consequently, this method could be a useful tool for preliminary evaluations and use in de-coupled conjugate studies.

**Author Contributions:** M.C.: Research and conceptualisation, methodology, experimental and numerical work and writing manuscript; A.M.: methodolody, review and editing; B.C.: methodology; P.I.: Supervisor; I.M.: Supervisor. All authors have read and agreed to the published version of the manuscript.

**Funding:** This research was funded by the EPSRC Transpiration Cooling grant (EP/P000878/1).

**Acknowledgments:** The authors wish to express their thanks for the on-going support provided by Rolls-Royce plc, and the EPSRC Transpiration Cooling grant (EP/P000878/1). The authors would also like to thank the technicians and staff at the Oxford Thermofluids Institute.

**Conflicts of Interest:** The funders had no role in the design of the study; in the collection, analyses, or interpretation of data; or in the writing of the manuscript.

## Nomenclature

The following abbreviations are used in this manuscript:

| | | |
|---|---|---|
| *Acronyms* | | |
| IR | Infrared | |
| OTI | Oxford Thermofluids Institute | |
| *Non-Dimensional Numbers* | | |
| $\eta_f$ | Film Effectiveness | |
| $\phi$ | Porosity | |
| $c_1$ | Spanwise distance constant | |
| $c_2$ | Spanwise shaping constant | |
| $M$ | Blowing ratio | |
| $m^*$ | Non-dimensional mass flow | |
| $x_{decay}$ | Streamwise distance film-decay | |
| *Properties* | | |
| $\alpha_t$ | Turbulent thermal diffusivity | $m^2/s$ |
| $\dot{m}$ | Mass flow rate | kg/s |
| $\gamma$ | Hole inclination angle | deg |
| $\lambda$ | Thermal conductivity | W/m·K |
| $\rho$ | Density | $kg/m^3$ |
| $A_f$ | Hole surface area | $m^2$ |
| $A_s$ | Surface coverage of film holes | $m^2$ |
| $c_p$ | Specific heat capacity | J/kg·K |
| $D$ | Diameter | m |
| $H$ | Plate thickness | m |
| $h$ | Heat transfer coefficient | $W/m^2$·K |
| $k$ | Thermal conductivity | W/m·K |
| $S_x$ | Streamwise spacing | m |
| $S_y$ | Spanwise Spacing | m |
| $T$ | Temperature | K |
| $V$ | Velocity | m/s |
| *Subscripts* | | |
| $\infty$ | Free stream | |
| aw | Adiabtic wall | |
| c | coolant | |
| f | film | |

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
