# Peer review of "Influence of Spanwise and Streamwise Film Hole Spacing on Adiabatic Film Effectiveness for Effusion-Cooled Gas Turbine Blades"

_ijtpp, doi:10.3390/ijtpp6030037_

Round 1

Reviewer 1 Report

Mandatory Request Changes:Mandatory Changes: Requested changes which are essential for the understanding and completeness of the paper. Paper of author(s) who have not complied with these requests may be rejected.:
No mandatory requests.

Recommended Requested Changes:Recommended Changes: Changes will improve the quality of the paper. Authors are strongly encouraged to comply with these requests.:
1. Please provide the main quantitative results to the “Abstract” and “Conclusion”.
2. Please create the section “Research object” and place it between “Introduction” and “Experimental Method”.
3. Please create section “Numerical Method” and place it before section “Experimental method”. Provide some details about modeling parameters (mesh, turbulence, model etc.).
4. Please remove extra line skip on the page 10.
5. Please check the consequence of the figure numbering. The reference to the figure 14 occur later than the references to the figures 15 and 16.

Author Response

  1. Please provide the main quantitative results to the “Abstract” and “Conclusion”.
    Some quantitative results now provided
  2. Please create the section “Research object” and place it between “Introduction” and “Experimental Method”.
    Done
  3. Please create section “Numerical Method” and place it before section “Experimental method”. Provide some details about modeling parameters (mesh, turbulence, model etc.).
    Empirical data used for Numerical method constants. Reference to this added. 
  4. Please remove extra line skip on the page 10.
    Done
  5. Please check the consequence of the figure numbering. The reference to the figure 14 occur later than the references to the figures 15 and 16.
    Done

Reviewer 2 Report

A paper with some very interesting results that would lead to discussion at the conference. I feel the paper is close to journal quality, but the following ‘mandatory changes’ must be made (or addressed) before making the decision. Also, the ‘recommended changes’ would, in this reviewers opinion, further improve the paper.

Mandatory changes
The authors categorise their film cooling work as ‘effusion cooling’ in the paper and include the term in the keywords section; however, it isn’t mentioned in the title or the abstract. It is suggested that effusion cooling be explicitly referred to in the abstract, and if possible, added to the title.

The term ‘superposition’ may not be familiar to all readers. It should be clearly defined as a concept when first introduced.

Add undefined symbols in Eqs. 2 and 3 to the nomenclature section.

For consistency, Eq. 2 should be modified to be a function of x and y, i.e. as it is written in Figure 1.

The locations of the holes in the individual film plot (upper diagram) and the
superpositioned plot (lower diagram) do not align in Figure 1. Please amend.

There are undefined geometric properties in Figure 2: l1, l2, l3 and l4. Please provide values for these in the supporting table. Also, provide dimensions for IR view area.

Change 1.12x104 to 1.12×104 on page 4 (i.e. replace lowercase x with multiplication symbol).

The ‘film fitting method’ is referred to at the bottom of page 4. Please provide details of this method to the reader.

‘Df’ is shown in Figures 3 and 4. Please define in the nomenclature section.

Figure 5 is mentioned before Figure 4 in the text. Re-order figures so they are in the order in which they are discussed.

Define porosity mathematically in terms of key parameters, including Sx, Sy and D.

Typographical error on line 6 of page 6. Change “the films become more isolated in, and distinct” to “the films become more isolated, and distinct”.

Please reword the sentence “However, the columns with no missing holes had a slower increase in film effectiveness” at the top of page 6 as it is currently a bit unclear.

Figure 9 needs to be larger – it was difficult to distinguish some of the features to which you referred in the text.

“Sy/D=Sy/D=3.0” on page 8 should be “Sx/D=Sy/D=3.0”. Please correct.

The experimental results section contains excellent findings, but the text is difficult to follow at times. I suggest making it clearer and more concise so that your discussion is easier for the reader to follow. Also, it is unclear why Figures 15 to 17 have been put in an Appendix when they form a key part of the discussion – I suggest moving them into the main report.

The last sentence on page 8 refers to an average film effectiveness difference of 0.6 – it appears to be lower than this in the plots (between 0.4 to 0.5). It follows that the rest of the paragraph is incorrect. Please comment on this observation.

Change the sentence “the film effectiveness is very encouraging for Sy/D=6.0” to “the film effectiveness is well-matched for Sy/D=6.0” (or similar).

Recommended changes
Change “The modified Goldstein-Sellars method showed relative agreement” to “The modified Goldstein-Sellars method showed relatively good agreement” in the abstract.

Change units for cp in the nomenclature to standard international units, i.e. from kJ/kg to J/kg.

The sentence starting “For temperatures to approach 2000K…” at the top of page 2 should be reworded as it does not read well.

Remove hyphens from the words ‘transpiration-’ and ‘effusion-’ on page 2 (midway through the page).

Explain what you mean by a ‘decoupled solver’ in the final paragraph of page 2.

The final paragraph of the introduction contains too much information on the geometries at this point. It is recommended that this is moved to the a dedicated ‘Geometries’ subsection in the ‘Experimental Method’ section. Also, it would be much clearer to the reader if geometric details were tabulated.

It is typical to have a paragraph at the end of the introduction providing an overview of each section of the paper – this would be helpful to the reader.

At the bottom of page 3 it is stated that “effort was made to reduce impact that the roughness would have”. Please clarify what efforts were made.

Remove hyphens from the words ‘streamwise-’ and ‘spanwise-’ on page 4 (bottom of the page).

Comparison area is difficult to see in Figures 3 and 4. Perhaps label for clarity.

Figures 7 and 8 start the ‘TD2 Results” section. Typically, sections are started with text – I suggest changing this.

Label TD2 on Figure 7 (or indicate in caption that TD2 is identified by the Sx/Sy=1 data).

State that TD2 is the central plot in the caption for Figure 7.

I suggest changing “approximately m*=1.81” to “approximately m*=1.8” at the bottom of page 7.

Author Response

A paper with some very interesting results that would lead to discussion at the conference. I feel the paper is close to journal quality, but the following ‘mandatory changes’ must be made (or addressed) before making the decision. Also, the ‘recommended changes’ would, in this reviewers opinion, further improve the paper.

Mandatory changes

The authors categorise their film cooling work as ‘effusion cooling’ in the paper andinclude the term in the keywords section; however, it isn’t mentioned in the title or the abstract. It is suggested that effusion cooling be explicitly referred to in the abstract, and if possible, added to the title.
Done

The term ‘superposition’ may not be familiar to all readers. It should be clearly defined as a concept when first introduced.
Explanation added to indicate the complimenting of films with subsequent downstream holes

Add undefined symbols in Eqs. 2 and 3 to the nomenclature section.
Done

For consistency, Eq. 2 should be modified to be a function of x and y, i.e. as it is written in Figure 1.
Done

The locations of the holes in the individual film plot (upper diagram) and the superpositioned plot (lower diagram) do not align in Figure 1. Please amend.
Done

There are undefined geometric properties in Figure 2: l1, l2, l3 and l4. Please provide values for these in the supporting table. Also, provide dimensions for IR view area.
Done

Change 1.12x104 to 1.12×104 on page 4 (i.e. replace lowercase x with multiplication symbol).
The ‘film fitting method’ is referred to at the bottom of page 4. Please provide details of this method to the reader.
This is the numerical method (Modified Goldstein-Sellers) as described previously. This has now been clarified with the following statement: ‘ …  the modified Goldstein method with Sellers superposition was implemented across a standardised geometry.’

‘Df’ is shown in Figures 3 and 4. Please define in the nomenclature section.s
All instances of Df changed to D for consistency.

Figure 5 is mentioned before Figure 4 in the text. Re-order figures so they are in the order in which they are discussed.
Discussion modified to keep figure layout. This change was taken to help with the presentation of the paper and the clarity of discussion.

Define porosity mathematically in terms of key parameters, including Sx, Sy and D.
Done

Typographical error on line 6 of page 6. Change “the films become more isolated in, and distinct” to “the films become more isolated, and distinct”.
Done

Please reword the sentence “However, the columns with no missing holes had a slower increase in film effectiveness” at the top of page 6 as it is currently a bit unclear.
Done

Figure 9 needs to be larger – it was difficult to distinguish some of the features to which you referred in the text.
Done

“Sy/D=Sy/D=3.0” on page 8 should be “Sx/D=Sy/D=3.0”. Please correct.
Done

The experimental results section contains excellent findings, but the text is difficult to follow at times. I suggest making it clearer and more concise so that your discussion is easier for the reader to follow. Also, it is unclear why Figures 15 to 17 have been put in an Appendix when they form a key part of the discussion – I suggest moving them into the main report.
Figures amended and discussion altered

The last sentence on page 8 refers to an average film effectiveness difference of 0.6 – it appears to be lower than this in the plots (between 0.4 to 0.5). It follows that the rest of the paragraph is incorrect. Please comment on this observation.
Updated to be 0.4 between peaks and troughs. Values for the rest of the paragraph are 0.2 and 0.15. Paragraph has been updated to reflect this.

Change the sentence “the film effectiveness is very encouraging for Sy/D=6.0” to “the film effectiveness is well-matched for Sy/D=6.0” (or similar).
Done

Recommended changes

Change “The modified Goldstein-Sellars method showed relative agreement” to “The modified Goldstein-Sellars method showed relatively good agreement” in the abstract.
Done

Change units for cp in the nomenclature to standard international units, i.e. from kJ/kg to J/kg.
Done

The sentence starting “For temperatures to approach 2000K…” at the top of page 2 should be reworded as it does not read well.
Done

Remove hyphens from the words ‘transpiration-’ and ‘effusion-’ on page 2 (midway through the page).
Done

Explain what you mean by a ‘decoupled solver’ in the final paragraph of page 2.
Done

The final paragraph of the introduction contains too much information on the geometries at this point. It is recommended that this is moved to the a dedicated ‘Geometries’ subsection in the ‘Experimental Method’ section. Also, it would be much clearer to the reader if geometric details were tabulated.
Done

It is typical to have a paragraph at the end of the introduction providing an overview of each section of the paper – this would be helpful to the reader.
Done

At the bottom of page 3 it is stated that “effort was made to reduce impact that the roughness would have”. Please clarify what efforts were made.
Done – update to high grade rohacell (closed-cell foam)

Remove hyphens from the words ‘streamwise-’ and ‘spanwise-’ on page 4 (bottom of the page).
Done
Comparison area is difficult to see in Figures 3 and 4. Perhaps label for clarity.

Figures 7 and 8 start the ‘TD2 Results” section. Typically, sections are started with text – I suggest changing this.
Done

Label TD2 on Figure 7 (or indicate in caption that TD2 is identified by the Sx/Sy=1 data).
Done

State that TD2 is the central plot in the caption for Figure 7.
Done

I suggest changing “approximately m*=1.81” to “approximately m*=1.8” at the bottom of page 7.
Done

Reviewer 3 Report

Mandatory Request Changes:Mandatory Changes: Requested changes which are essential for the understanding and completeness of the paper. Paper of author(s) who have not complied with these requests may be rejected.:
The work could be a preliminary study for future investigations related to the extension of the Sellers superposition method. The results are in line with expectations but some critical issues arise both from the experimental and numerical side:
1) The test rig exhibits a strong non-uniformity in the spanwise direction, probably due to the impact of lateral boundaries or the coolant/mainstream mass flow ratio: this issue could be better discussed.
2) Numerical simulations and experiments present some differences regarding the operating conditions (for example the coolant mass flow): could it be fixed?

Recommended Requested Changes:Recommended Changes: Changes will improve the quality of the paper. Authors are strongly encouraged to comply with these requests.:
Detailed comments in the attached file.

Author Response

 The work could be a preliminary study for future investigations related to the extension of the Sellers superposition method. The results are in line with expectations but some critical issues arise both from the experimental and numerical side:

1) The test rig exhibits a strong non-uniformity in the spanwise direction, probably due to the impact of lateral boundaries or the coolant/mainstream mass flow ratio: this issue could be better discussed.

Upon further review of results and some other full domain CFD previously done at the lab we believe this is a result of the coolant acting as a body that the mainstream must move around. Near the lateral boundaries the flow can move towards one side as well as upwards, however centrally the flow can only curve upwards. Consequently, the central cooling holes experience a higher static pressure, and thus the mass flow is reduced. As cooling hole spacing is increased, the mainstream flow has opportunity to go between the hole rows, hence the central holes are subject to a lower pressure and therefore less restriction on the mass flow is imposed.

2) Numerical simulations and experiments present some differences regarding the operating conditions (for example the coolant mass flow):

could it be fixed?

Fixed: Numerical study updated to correspond with experiments, and this is reflected with the updated results.

Round 2

Reviewer 1 Report

Article could be accepted in the present form.

Author Response

Final manuscript to be submitted

Reviewer 2 Report

Most changes made from first review comments (except "Comparison area is difficult to see in Figures 3 and 4. Perhaps label for clarity") - this is now journal quality. Please see the attached document for final changes/suggestions

Author Response

1) 'L' dimension added in figure and x and y in table corrected to format 

2) Clarification that in this part of the research the numerical approach (modified Goldstein-Sellers) was used 

3) Table 2 position moved

4) lines 141-142 comma added 

5) Figure 4 dimensions amended 

6) Changes to figure 5 made. For clarity of streamwise length (x dimension) the number of hole rows with spacing and overall length has been added ie nSx = 4 (6D) = 30D. 

7) Statement on line 206 and 207 corrected 

8) Further analysis for error between Numerical and Experimental work is ongoing for further studies. 

9) Section titles changed: Sec 3 = Results and Discussion, Sec 4 = Conclusions